# Effect of Hydrothermal Factors on the Microhardness of Bulk-Fill and Nanohybrid Composites

**DOI:** 10.3390/ma16052130

**Published:** 2023-03-06

**Authors:** Daniel Pieniak, Agata M. Niewczas, Konrad Pikuła, Leszek Gil, Aneta Krzyzak, Krzysztof Przystupa, Paweł Kordos, Orest Kochan

**Affiliations:** 1Tribology Center, Łukasiewicz Research Network-Institute for Sustainable Technologies (L-ITEE), Ul. Pułaskiego 6/10, 26-600 Radom, Poland; 2Faculty of Transport and Computer Science, WSEI University, Projektowa 4, 20-209 Lublin, Poland; 3Department of Conservative Dentistry with Endodontics, Medical University of Lublin, W. Chodźki 6, 20-093 Lublin, Poland; 4Faculty of Aeronautics, Military University of Aviation in Dęblin, 35 Dywizjonu 303, 08-521 Deblin, Poland; 5Department of Automation, Lublin University of Technology, Nadbystrzycka 36, 20-618 Lublin, Poland; 6Institute of Transport, Combustion Engines and Ecology, Lublin University of Technology, Nadbystrzycka 36, 20-618 Lublin, Poland; 7School of Computer Science, Hubei University of Technology, Wuhan 430068, China; 8Department of Measuring Information Technologies, Lviv Polytechnic National University, Bandery Str. 12, 79013 Lviv, Ukraine

**Keywords:** bulk-fill dental composites, microhardness, thermocycling, hydrothermal degradation

## Abstract

This study evaluates the effect of aging in artificial saliva and thermal shocks on the microhardness of the bulk-fill composite compared to the nanohybrid composite. Two commercial composites, Filtek Z550 (3M ESPE) (Z550) and Filtek Bulk-Fill (3M ESPE) (B-F), were tested. The samples were exposed to artificial saliva (AS) for one month (control group). Then, 50% of the samples from each composite were subjected to thermal cycling (temperature range: 5–55 °C, cycle time: 30 s, number of cycles: 10,000) and another 50% were put back into the laboratory incubator for another 25 months of aging in artificial saliva. The samples’ microhardness was measured using the Knoop method after each stage of conditioning (after 1 month, after 10,000 thermocycles, after another 25 months of aging). The two composites in the control group differed considerably in hardness (HK = 89 for Z550, HK = 61 for B-F). After thermocycling, the microhardness decrease was for Z550 approximately 22–24% and for B-F approximately 12–15%. Hardness after 26 months of aging decreased for Z550 (approximately 3–5%) and B-F (15–17%). B-F had a significantly lower initial hardness than Z550, but it showed an approximately 10% lower relative reduction in hardness.

## 1. Introduction

Modern conservative dentistry makes use of improved restorative materials. Studies have been conducted to develop composite materials with a multi-component structure. One of the products of such research is the modern ceramic–polymer composite, and materials classified as Polymer Matrix Ceramic Composites (PMCCs) deserve special attention [1]. PMCCs are also often referred to as resin-based composites (RBCs) [2]. Nowadays, the RBC technology is one of the most intensively developing technologies in the field of dental materials. RBCs are used in a wide variety of clinical applications [3]. Such composites were created thanks to the development of dental materials. Their structure also evolves [4]. Currently, new types of these composites, differing mainly in the structure of powder fillers, are being developed. They can be divided into microfilled, nanofilled, and nanohybrid composites. Hybrid composites usually contain 70–80% of a glass-based filler and sometimes 20–30% of nanofillers [5]. For quite a long time, the additives, such as nano-sized zirconium, titanium and aluminum oxide particles, have been used [6,7]. A particularly promising type of RBCs are bulk-fill composites. In recent years, bulk-fill composites have become a standard of modern clinical practice [8]. They have been designed to improve the efficiency of the treatment of carious lesions in side teeth which need to withstand the largest mechanical loads. The ability of a material to withstand high mechanical loads without sustaining fatigue damage is crucial from the point of view of clinical applications [9,10]. However, the existing literature reports that the mechanical properties of bulk-fill composites are incomplete and difficult to compare [11,12]. Both patients and dentists expect a dental material to maintain its required mechanical properties for a long time in the difficult conditions of the oral cavity. In the mouth, a restorative material is in constant contact with saliva, which leads to hydrolytic degradation of the filling [13]. The mechanism of hydrolytic degradation is not fully understood. It is known that RBCs absorb moisture from surroundings, and chemical compounds in the aqueous solution decompose the RBC polymer network by breaking down ester bonds [14]. It is also believed that the level of moisture absorption depends on the quantity and type of filler particles and the type of monomers that build the matrix [15]. Additionally, there are factors in the mouth that lead to thermal fatigue, which works simultaneously with the aging processes. The oral cavity is usually characterized by changing thermal conditions, both steady-state and non-steady-state. This can lead to thermal stresses. A point on the body in which the temperature does not change with time is referred to as a steady-state field; all other points are non-steady state fields [16]. RBC restorations can have both types of fields. When the temperature of the oral cavity environment is constant, its value is close to the physiological temperature range of 36.3–37.1 °C for men and 36.5–37.3 °C for women [17]. During food consumption, the oral temperature changes due to heat transfer (non-steady-state conditions). Some authors report that there are 50,000 thermal cycles in the mouth per year [18]. In laboratory conditions, a simulated [19] thermal fatigue process is referred to as thermocycling [20,21]. Thermocycling is a process of simulating fluctuations in actual temperature by recreating, for e.g., physiological processes such as eating, drinking, or breathing [13]. There are now no universally accepted standards for temperature variability in thermocycling [22]. The thermocycling temperature range is usually from 5 °C to 55 °C or even 60 °C [23,24,25]. The degradation of polymer composites is usually irreversible and leads to the deterioration of the mechanical properties of these materials, including their microhardness. Microhardness is currently used to evaluate properties of the surface layer of dental composites [26]. Microhardness of dental composites for fillings is related to the natural properties of human teeth [27,28]. In this study, according to the assumptions in [29,30,31], changes in the microhardness of the surface layer of polymer composite specimens were measured as indicators of the decrease in resistance to the mechanical wear of the composites. The aim of this study is to evaluate the effect of aging under the influence of artificial saliva and thermal shocks on the microhardness of a bulk-fill composite compared to a nanohybrid composite.

According to the manufacturer, Filtek Z550 (3M ESPE) (henceforth abbreviated as Z550) displays high compressive, bending, and tensile strengths as well as good abrasion resistance [32], whereas Filtek Bulk-Fill (3M ESPE) (henceforth abbreviated as B-F) has good compressive and bending strengths [33] and good abrasion resistance [34].

The two composites exhibit a similar wear resistance (as declared by their manufacturers [32,33,34]). Considering these manufacturer-declared parameters, and taking into account the similarity in the wear resistance of the two materials, we formulated the following four research hypotheses:The initial mechanical properties of the surface of the tested materials will be similar. Due to the similar wear resistance, the differences in hardness will be small (<5% of their average value).Cyclic thermal loading will cause a significant decrease in the microhardness of both test materials.The mean decrease in the microhardness of Z550 and B-F, exposed to the same thermal shocks, will be statistically comparable (similar).Twenty-six months of aging in a humid environment in the temperature of 37 °C will cause a significant hardness deterioration of the tested materials.

## 2. Materials and Methods

### 2.1. Material

The composites Z550 and B-F are commercially available materials. Z550 contains clusters of nanoparticles of zirconium and silicon compounds as well as nanosilica particles [34]. B-F is used in the reconstruction of posterior teeth [33]. Properties of these materials are summarized in Table 1.

### 2.2. Method

#### 2.2.1. Sample Preparation

The nominal dimensions of disc-shaped samples, according to ISO 4049 [35], were set as follows: a diameter of 15 mm, a thickness of 1 mm (Figure 1). The material was laid in layers of 1 mm in the mold, according to the manufacturer’s recommendations. The top surface was then irradiated with a Megalux LED lamp (Megadenta, Radeberg, Germany) at 1200 mW/cm^2^ for 40 s using a soft start system (Figure 1). After curing and marking the irradiated side (black dot at the edge of the sample shown in Figure 1), the samples were removed from the mold and placed in containers with artificial saliva without further processing. Samples of each material consist of 10 specimens (in total N = 20 specimens).

#### 2.2.2. Aging and Thermocycling

Specimens were immersed in containers with artificial saliva (AS) (pH = 5.3) and aged for 1 month in a Q-Cell laboratory thermal chamber (Pol-Lab, Wilkowice, Poland) at 37 °C. The composition of artificial saliva (AS) was as defined using the PN-EN ISO 10271:2012 standard [36]. The composition of artificial saliva according to this standard is as follows: 0.4 g NaCl, 0.4 g KCl, 0.795 g CaCl_2_·H_2_O, 0.78 g NaH_2_PO_4_·H_2_O, 0.005 g Na_2_S·9H_2_O, 1 g of urea, 1000 mL of distilled water, pH: 5.2–5.5. Then, the microhardness measurements for all samples were performed. After the microhardness measurements, half of the samples of each material (n_Z550_ = 5, n_B-F_ = 5) were placed back in the thermal chamber. The total time spent in the chamber was (1 month + 25 months) 26 months. The other half of the samples (n_Z550_ = 5, n_B-F_ = 5) were thermocycled.

The simulations were performed with the use of a thermal shock simulator designed by the authors [37]. The simulator consists of a hydraulic unit and a microprocessor control system. The simulator induced abrupt changes in the temperature of the liquid (water, in this particular test) in the measuring vessel, into which the specimens were immersed (Figure 1). The measuring vessel was filled alternately with the heated (55 °C) or cooled (5 °C) working liquid with two separate pumping systems. One thermocycle lasted 201 s and included pumping the cooled liquid into the vessel with the specimens (35 s), holding the cooled liquid in the vessel—t_min_ (30 s), pumping out the cooled liquid (35 s), pause (0.5 s), pumping the heated liquid into the specimen vessel (35 s), holding the heated liquid in the vessel—t_max_ (30 s), pumping out the heated liquid (35 s), and pause (0.5 s). The microhardness measurements for the samples after thermocycling were carried out again. Then, after the aging in artificial saliva for 26 months, the microhardness measurements of the aged specimens were carried out again.

#### 2.2.3. Microhardness Test

Microhardness was measured using the Knoop method [38,39]. In the Knoop hardness test, the tested material is statically indented with a pyramidal diamond point with the face angles of 172°30′ for the long edge and 130° for the short one (Figure 2). Hardness measured using this method is proportional to the ratio of load to the area of permanent impression made by the indenter. It is calculated from the following formula:(1)$$HK=14.228\frac{0.102P}{L^{2}},$$
where: *P*—load [N], *L*—length of indentation along its longer axis [mm].

Microhardness tests were run using disk-shaped specimens, according to the ISO4049 standard (Figure 2). Knoop microhardness was measured using the Futuretech FM 700 device (Future-tech Corp., Kawasaki, Japan) under the load of 0.025 kg. Indenter penetration time was set to 25 s. The tests were carried out on the light-cured (LC, top side of a sample) and non-light-cured (NLC, bottom side of a sample) surfaces of the specimens (Figure 2).

#### 2.2.4. Statistical Analysis

The data were statistically analyzed using the Statistica software. The histograms of numbers were plotted and descriptive statistics were calculated. The normality of the distribution was assessed using the Shapiro–Wilk test. The Wilcoxon signed-rank test was used to compare the groups. The Wilcoxon test, which is a non-parametric equivalent of the Student’s parametric t-test, was used. The Wilcoxon test consists of ranking differences in measurements between test groups. The relative decrease in the microhardness of a material on its LC surface was calculated from the following formula:$$D=1-\frac{H}{H_{0}}$$
where: *D*—relative decrease in microhardness, *H*—mean microhardness after thermocycling or after 26 months aging in AF, *H*_0_—mean microhardness in the control group.

## 3. Results

The results of the microhardness measurements are summarized in the histograms (Figure 3, Figure 4, Figure 5, Figure 6, Figure 7 and Figure 8). These are plots to assess the empirical distribution of the measurement results. Moreover, such a plot facilitates the assessment of the normality of empirical distribution because the adjusted probability density curves of the normal distribution are superimposed on the histogram. The plots show cardinality (N), percentages, and descriptive statistics (StdDv—standard deviation, Max—maximum value, Min—minimum value). The analysis focused on whether the distributions of the measurement results are normal. The Shapiro–Wilk (S-W) test was applied to check whether the distribution of measurement results is normal. The null hypothesis of the S–W test assumes that a specimen is derived from the normally distributes population. If the S–W test is statistically significant (*p* < 0.05), the statistical distribution of the specimens is not normal. For Z550, five out of six groups showed *p* < 0.05 (Z550: LC *p* = 0.0402; NLC *p* = 0.0000; LC 10^3^ TC *p* = 0.3046; NLC 10^3^ TC *p* = 0.0236; LC 26 mths_37C *p* = 0.0061; NLC 26 mths_37C *p* = 0.0111), whereas for B-F, three out of six groups showed *p* < 0.05 (B-F: LC *p* = 0.0003; NLC *p* = 0.1949; LC 10^3^ TC *p* = 0.0366; NLC 10^3^ TC *p* = 0.1076; LC 26 mths_37 °C *p* = 0.0001; NLC 26 mths_37 °C *p* = 0.0732). The mean of the hardness measurements of the composites in the control group is not the same and differs by about 28 Knoop hardness units (Figure 3 and Figure 4), or the mean hardness of the light-cured surface of B-F was lower by about 31% than that of Z550 (LC). The differences in the hardness of the non-light-cured surfaces of both composites (NLC) were lower, and the mean hardness of B-F compared to that of Z550 was less by about 26%. The statistical significance of the differences between the groups of the results was confirmed using the Wicoxon test. In our study, the number of measurements in each group was larger than 25. In such cases, the Z-statistic is computed, which stands for the highest rank sum. A significant test result (*p* < 0.05) indicated that one of the measurements had a much higher value than the other (s). The differences between the materials and the differences between the LC and NLC groups were evaluated for each material. The *p*-values, indicating statistically significant differences, are marked red in Table 2. Statistically significant differences in the hardness of the materials were demonstrated.

This means that the first assumption of the hypothesis should be considered unconfirmed.

Next, the second assumption of the hypothesis was tested. We analyzed the difference between the hardness of the specimens that had been kept in artificial saliva for one month and that of the specimens exposed to thermal shocks (Figure 3, Figure 4, Figure 5 and Figure 6). The mean microhardness of most of the tested materials deteriorated after 10,000 hydro-thermal cycles (Figure 5 and Figure 6). After thermocycling, the mean hardness dropped, and the decrease was greater for Z550 (approx. 22–24%) than for B-F (approx. 12–15%). These differences were smaller than in the analysis of the first assumption of the hypothesis, especially for the B-F composite. The Wilcoxon matched pair test showed that the hardness test results for the specimens exposed to thermal shocks differed significantly from those obtained for the control ones (Table 3). The histograms, descriptive statistics, and Wilcoxon test results supports the second assumption of the hypothesis.

As a third step in the statistical analysis of the measurement results, we tested the significance of differences in hardness decreases due to fatigue in the test materials (Table 4). The highest degradation parameter D was noticed for the light-cured surface of Z550. This parameter was lowest for the light-cured surface of B-F. The D parameters for the light and non-light cured surfaces of both composites are similar, which is clear from the results in Table 2 (two last lines—5, 6).

The test results demonstrated (Figure 7 and Figure 8) that the two materials showed a different degree of degradation. A higher relative decrease in hardness (approx. 22–24%) was observed for Z550 (Figure 7), whereas the hardness of B-F dropped by 12–15 percent (Figure 8).

The fourth considered hypothesis involves the effect of aging for 26 months in a humid environment at the temperature of 37 °C (hydro-isothermal influence). The results after one month of aging were compared with the results of microhardness after an additional 25 months of aging. The descriptive statistics and histograms indicate a decrease in microhardness. The statistical Wilcoxon test was applied again to check statistical significance (Table 5). The test results confirm the significant differences for the B-F material, but no statistically significant differences were noticed for the Z550 material.

The degree of degradation, D, was also calculated from the previous formula, and the results are given in Table 6. The highest D parameter was for the NLC surface of the B-F material, whereas its lowest value was for the LC surface of the Z550 material. For the NLC surfaces, the degradation parameter was slightly higher. The degree of degradation of the Z550 material is significantly lower than in thermocycling (hydro-anisothermal), whereas it is similar after thermocycling for the B-F composite.

## 4. Discussion

Loads in the oral cavity lead to stresses in dental fillings [40]. These are contact stresses that arise during mastication as a result of a redistribution of normal and tangential occlusal forces through the intermediate layer, such as when food is being chewed, or more precisely, when a suspension of food particles and wear products happens [41]. Occlusal forces vary individually but can be as high as 1000 N [42], and since the contact area between the opposing teeth is approx. 0.4–2.2 mm^2^ [43], the stresses under these loading conditions can reach 0.45–2.5 GPa [42]. The highest forces are observed in the top layer of a composite restoration, so it is fundamental to evaluate microhardness if dental composites are to be evaluated in terms of their use. It is known from the literature that microhardness tests allow us to indirectly assess some clinically relevant properties of a composite. As demonstrated experimentally [44], there is a correlation between a composite’s microhardness and the amount of polymerization shrinkage. It has also been shown that the wear of composite fillings is related to their microhardness under in vitro simulation conditions [45]. Microhardness tests can also be used to assess the local photopolymerization gradient [46]. The results of our research demonstrate that the difference in the initial microhardness of both materials is significant (Figure 3 and Figure 4). The highest initial hardness was recorded for the Z550 composite. It was found that hardness depends on the type and share of the filler in the structure. The dependence of hardness on the content of the filler in the composite was demonstrated in [47,48]. The filler content in the structure of both composites is similar and slightly higher for Z550. It is claimed [49] that the share of filler particles has an influence on hardness but that the influence of the type and size of particles of a dispersed phase is greater. The size of particles of a dispersed phase can also be significant and, as it is stated in [50], the use of nanoparticles as part of a filler increases the hardness of the material. The degree of conversion (DC) is also correlated with hardness. It was shown that a strong positive correlation exists between Knoop microhardness and the DC [51,52]. The adhesion of the filler and the matrix is probably also fundamental. The differences can also be explained by the content of the Bis-GMA polymer resin that is harder after photopolymerization than the UDMA resin. These resins differ in terms of both degree of polymerization and molecular rigidity [53].

What is more, no significant differences were found in the hardness of the light-cured (LC) and non-light-cured (NLC) surfaces of the specimens. For the assumed thickness of the specimens, the differences were too statistically insignificant, and even for a thickness from 2 mm to 4 mm [54], no differences were noticed between the LC and NLC surfaces for certain dental composites. However, this study’s aim was not to test the depth of polymerization, as it was in [55], a study that showed the greatest decrease in nanohardness (via another test method, the Berkovich indenter test on a force (universal) hardness tester) below 1 mm from the light-cured surface of the half-cylinder. It can also be stated that according to the ISO 4049 standard [35], the depth of hardening of a single layer should not be less than 1.5 mm, and this depth is even greater for bulk-fill composites. Ilie [56] states that some brands of bulk-fill RBCs can be polymerized in up to 4-mm thick increments. It is claimed [57] that the real depth of hardening can be specified from the depth corresponding to 80% of the hardness of the light-cured surface. What is more, specimens of bulk-fill composites can be thicker than 4–6 mm. After curing during the time recommended by the manufacturer, if the ratio of the hardness of the lower surface (NLC) to the upper surface (LC) is higher than or equal to 80%, then the composite meets the requirements specified for bulk-fill materials. Another type of load in the oral cavity is thermal loads. Under the present experimental conditions, the test RBCs experienced thermal fatigue in the moist environment. This process can be thought of as being an effect of the convective flow of heat through the surface of a solid body which is in contact with a heated or cooled liquid (food suspension, hot or cold beverage). By and large, one can distinguish two mechanisms of convection: free convection—when the fluid moves as a result of a temperature difference alone, and forced convection—when fluid movement is forced, as is the case with the oral cavity. The action of viscous forces leads to the formation of a hydrodynamic boundary layer in the fluid near the surface of the solid washed by the fluid. When a solid, e.g., a composite filling, is washed by a fluid with a different temperature, a thermal boundary layer is formed near the surface of this body. Such a state can lead to the formation of thermal loads [31] due to an uneven temperature distribution or a change in body temperature when the support is statically indeterminate or the thermal expansion coefficient is non-uniform and can be treated as a case of loading [16]. Thermal loading of the surface leads to the formation of a temperature gradient in the material, accompanied by second-type thermal stresses. Moreover, in the case of multiphase materials, including powder-reinforced composites, the individual filler particles have different directions of thermal expansion. This results in the formation of first-type thermal stresses as well [16]. When the changes in temperature are cyclic and limited in time, the diffusion of heat is limited to the surface layer. This means that the surface layer of a material is characterized by the greatest variation in thermal stresses and therefore undergoes the highest risk of degradation. Microhardness as a mechanical property of the surface layer can justifiably be used in experimental assessments of the state of this layer [58]. In some studies, hardness has been described as a measure of the resistance of polymers to abrasive wear [45,59]. In general, abrasive wear depends on the hardness of a material in the friction node, and the shape and protruding irregularities of the harder component of a kinematic couple act as micro-blades. It is claimed [60] that the wear resistance of bulk-fill composites was similar to that of a conventional composite or even higher [61]; however, the bulk-fill composite turned out to be less hard in our research. It has been found that hardness, similarly to the elastic modulus, depends to a large extent on the composite’s matrix [62]. In other words, a decrease in microhardness caused by hydro-thermal degradation can be interpreted as the degradation of the matrix. A role might be played here by the softening of the polymer phase which depends on the presence of water, the temperature of the polymer, and the difference between the glass transition temperature and the ambient temperature [63,64]. For the tested materials, the content of the Bis-GMA resin in the Z550 structure may play an important role in the rate of degradation in a humid environment. The polar hydroxyl groups in Bis-GMA tend to form hydrogen bonds with water, which is hydrophilic macroscopically. The higher water sorption of the Bis-GMA-containing composite resulted from the stronger interactions of hydrogen bonding formed by hydroxyl [65]. It has to be noted that hardness is not dependent solely on the polymer matrix. The authors [62] observed that Vickers hardness increased almost linearly with the increasing filler content. This factor may have led to the differences in the hardness of the composites in the control group, because the Z550 composite had a slightly higher filler content. In our experiments, a drop in microhardness was recorded for both composites after thermocycling [63]. Decreases in the microhardness of materials subjected to thermal loads have also been observed by other authors, including [66] (temperature range: 5–55 °C, number of cycles: 10,000, exposure time: 30 s). Some other researchers, however, have demonstrated that the properties of the surface layer of a material can improve under the influence of cyclic thermal loading. For example, in [67], the hardness of the RBC material called Sinfony increased under the influence of thermal loads (temperature range: 5–55 °C, number of cycles: 5000, exposure time: 30 s). The authors of [24] observed an increase in the nanoindentation hardness of specimens subjected to thermal loading (temperature range: 5–55 °C, number of cycles: 2000, exposure time: 30 s) and immersed for 48 h in a liquid at a temperature of 55 ± 5 °C. In those experiments, the surface properties of the composites may have become better as a result of the improved bonding of the filler particles to the matrix and a reduction in voids in the material due to thermal stresses. Thermal stresses lead to the secondary polymerization of materials. It has been argued that due to the improved bonding and fewer voids in the material, the diffusion of water into the material is reduced [36]. However, the differences in findings may also be due to the different numbers of load cycles used and dissimilarities in some cycle parameters. They can also be explained by the fact that thermal fatigue is a process that occurs in stages, as already postulated in other publications not regarding multiphase materials, i.e., RBCs. According to [68], three stages of thermal fatigue can be distinguished: thermocyclic strengthening or weakening, repeated processes of strengthening and weakening, and destruction. It seems that the composites we tested here may have strengthened at the beginning of the process but then their surface strength begun to decrease. A behavior of this type was observed for some RBCs in [59]. The test materials, despite similarities in structure and some properties, differed in their resistance to the conditions of the oral cavity such as moisture and thermal cycling. Moisture and constant temperature (thermal influence) cause less intensive degradation than thermocycling (hydro-anisothermal influence).

## 5. Conclusions

Bulk-fill composites are used in dentistry to fill cavities in posterior teeth which carry the largest mechanical loads. It was shown that the microhardness of these composites declines as a result of cyclic hydro-thermal changes in the oral cavity.The bulk-fill composite was demonstrated to have a lower initial microhardness than Z550. At the same time, the relative decrease in the microhardness of the bulk-fill material B-F after thermal cycling was significantly lower than of Z550.The hydro-isothermal impact for 26 months caused a decrease in microhardness. For the nanohybrid composite, it was lower than the degree of degradation under hydro-anisothermal conditions (thermocycling), whereas for the bulk-fill composite, it was similar in both cases of hydro-thermal influence.

## Figures and Tables

**Figure 1 materials-16-02130-f001:**
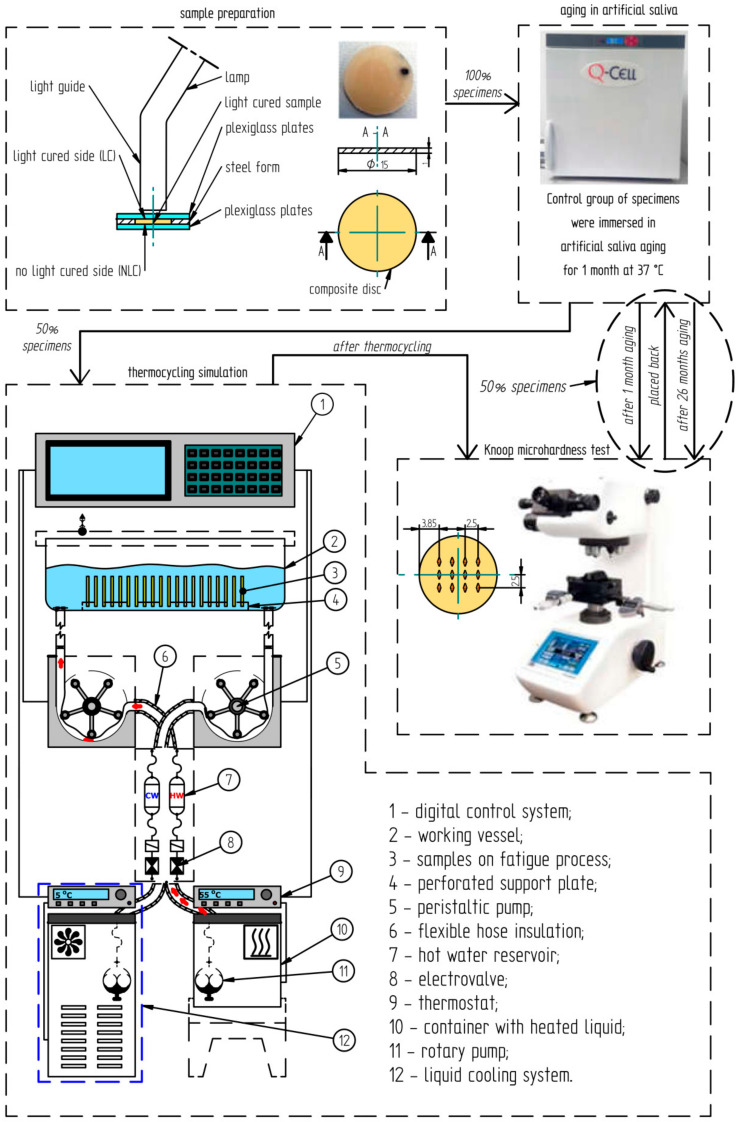
Schematic of the course of testing the dental composite samples.

**Figure 2 materials-16-02130-f002:**
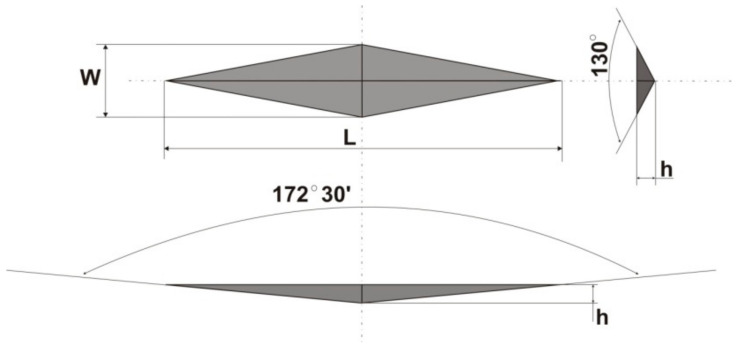
The image of an indentation left by the Knoop indenter.

**Figure 3 materials-16-02130-f003:**
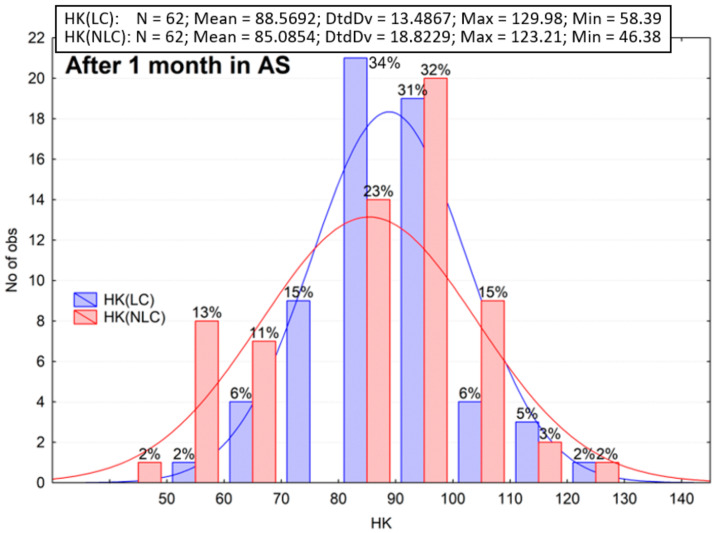
The results of Knoop hardness measurements for Z550 after aging in artificial saliva (AS) for one month. LC—light-cured surface of sample (top side of sample), NLC—non-light-cured surface (bottom side of sample), HK—Knoop hardness.

**Figure 4 materials-16-02130-f004:**
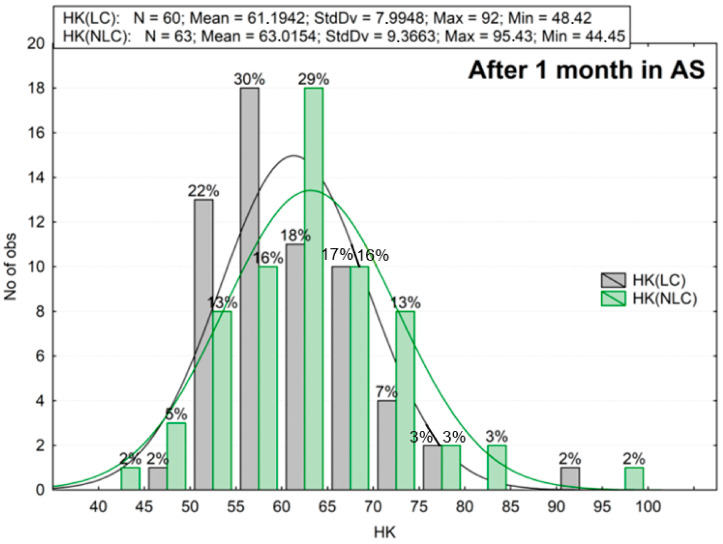
The results of Knoop hardness measurements for B-F after aging in artificial saliva (AS) for one month. LC—light-cured surface of sample (the top side of a sample), NLC—non-light-cured surface (the bottom side of a sample), HK—Knoop hardness.

**Figure 5 materials-16-02130-f005:**
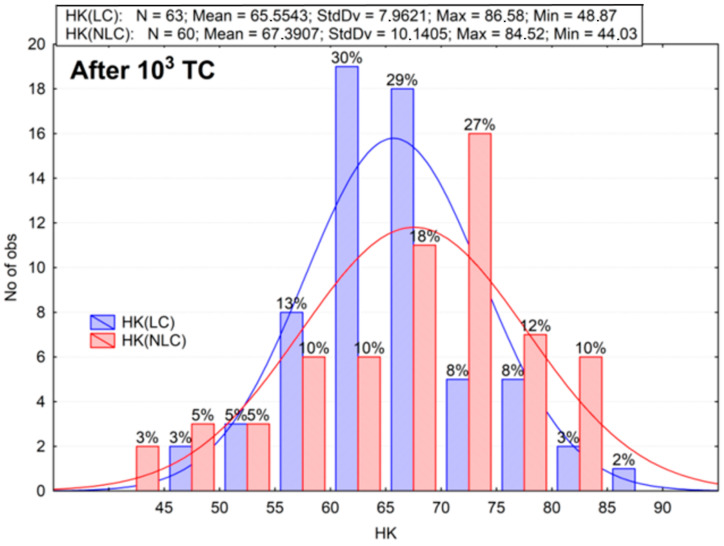
The results of Knoop hardness measurements for Z550 after thermocycling (TC). LC—light-cured surface of sample (top side of sample), NLC—non-light-cured surface (bottom side of sample), HK—Knoop hardness.

**Figure 6 materials-16-02130-f006:**
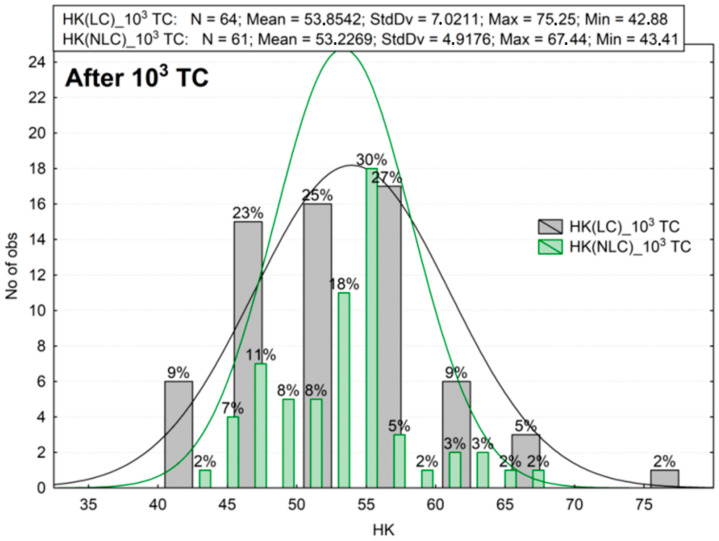
The results of Knoop hardness measurements for B-F after thermocycling (TC). LC—light-cured surface of sample (top side of sample), NLC—non-light-cured surface (bottom side of sample), HK—Knoop hardness.

**Figure 7 materials-16-02130-f007:**
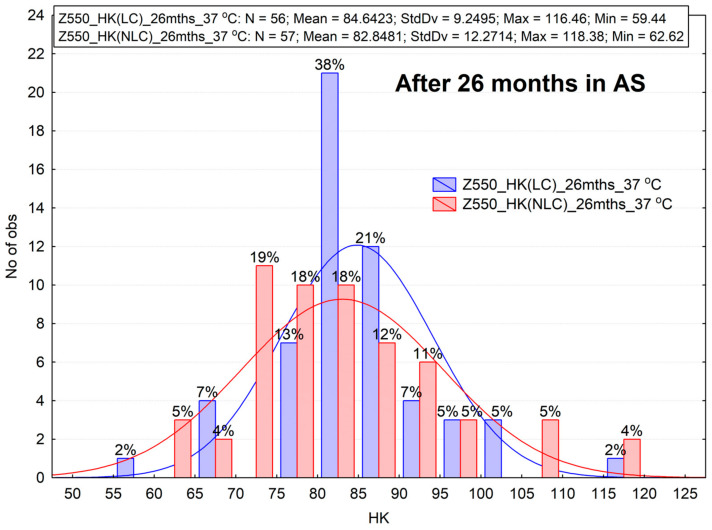
The results of Knoop hardness measurements for Z550 after aging in artificial saliva (AS) for 26 months. LC—light-cured surface of sample (the top side of a sample), NLC—non-light-cured surface (the bottom side of a sample), HK—Knoop hardness.

**Figure 8 materials-16-02130-f008:**
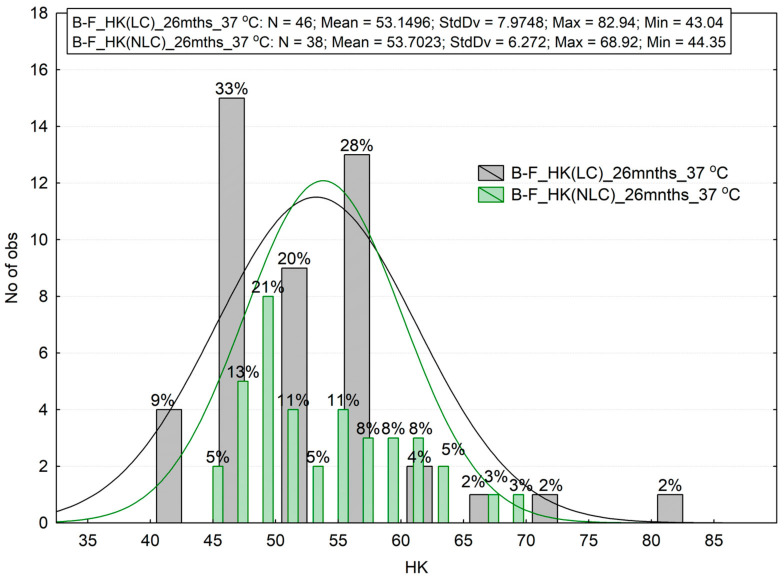
The Knoop hardness test results for B-F after aging in artificial saliva (AS) for 26 months. LC—light-cured surface of sample (the top side of a sample), NLC—non-light-cured surface (the bottom side of a sample), HK—Knoop hardness.

**Table 1 materials-16-02130-t001:** PMCCs (Polymer Matrix Ceramic Composites) used in this study and their composition and percentage fraction (weight [wt.%]) of inorganic fillers [33,34].

Material	Filtek Z550	Filtek Bulk-Fill
Manufactorer	3M ESPE	3M ESPE
Type	Universal restorative material	Posterior restorative material
Matrix	Bis-GMA, UDMA, Bis-EMA, PEGMA, TEGMA	AUDMA, UDMA and 1, 12-dodecane-DMA
Filler	SiO_2_ (particle of 20 nm), ZrO_2_/SiO_2_ (particle of 5–20 nm)	20 nm silica; 4 to 11 nm zirconia filler, zirconia/silica cluster filler (comprised of 20 nm silica and from 4 to 11 nm zirconia particles) ytterbium trifluoride filler consisting of agglomerate 100 nm particles
Content of filler particles (wt.%)	78.5	76.5

**Table 2 materials-16-02130-t002:** Wilcoxon test results. Significance of the differences in the microhardness of the tested materials in the control group. LC—light-cured surface of sample (top side of sample), NLC—non-light-cured surface (the bottom side of a sample), HK—Knoop hardness.

No.	Group vs. Group (Control)	Z	*p*-Level
1	Z550_HK(LC) & B-F_HK(LC)	6.350940	0.000000
2	Z550_HK(LC) & B-F_HK(NLC)	6.491959	0.000000
3	Z550_HK(NLC) & B-F_HK(LC)	5.909220	0.000000
4	Z550_HK(NLC) & B-F_HK(NLC)	5.873101	0.000000
5	Z550_HK(LC) & Z550_HK(NLC)	0.315832	0.752113
6	B-F_HK(LC) & B-F_HK(NLC)	1.424592	0.154276

**Table 3 materials-16-02130-t003:** Wilcoxon test results. The significance of the differences in the microhardness of the aged specimens for one month and the specimens exposed to thermocycling. LC—light-cured surface of sample (top side of sample), NLC—non-light-cured surface (bottom side of sample), HK—Knoop hardness.

No.	Group vs. Group (Control)	Z	*p*-Level
1	Z550_HK(LC)_10^3^ TC & Z550_HK(LC)	6.391308	0.000000
2	Z550_HK(LC)_10^3^ TC & Z550_HK(NLC)	5.856787	0.000000
3	Z550_HK(NLC)_10^3^ TC & Z550_HK(LC)	6.040933	0.000000
4	Z550_HK(NLC)_10^3^ TC & Z550_HK(NLC)	5.404613	0.000000
5	B-F_HK(LC)_10^3^ TC & B-F_HK(LC)	3.736833	0.000186
6	B-F_HK(LC)_10^3^ TC & B-F_HK(NLC)	5.152533	0.000000
7	B-F_HK(NLC)_10^3^ TC & B-F_HK(LC)	4.954163	0.000001
8	B-F_HK(NLC)_10^3^ TC & B-F_HK(NLC)	5.035507	0.000000

**Table 4 materials-16-02130-t004:** Changes in the D parameter of the test materials after thermocycling. LC—light-cured surface of sample (the top side of a sample), NLC—non-light-cured surface (the bottom side of a sample), HK—Knoop hardness.

D	H/H_0_	1-H/H_0_	Material	Specimen Plane
0.24735	0.75265	0.24735	Z550	LC
0.120093	0.879907	0.120093	B-F	LC
0.226805	0.773195	0.226805	Z550	NLC
0.155335	0.844665	0.155335	B-F	NLC

**Table 5 materials-16-02130-t005:** Wilcoxon test results. The significance of the differences in the microhardness of the aged specimens for one month and specimens aged for 26 months. LC—light-cured surface of sample (the top side of a sample), NLC—non-light-cured surface (the bottom side of a sample), HK—Knoop hardness.

No.	Control Group vs. Group	Z	*p*-Level
1	Z550_HK(LC) & Z550_HK(LC)_26mths_37 °C	0.699863	0.484014
2	Z550_HK(LC) & Z550_HK(NLC)_26mths_37 °C	1.329610	0.183648
3	Z550_HK(LC) & B-F_HK(LC)_26mths_37 °C	5.645404	0.000000
4	Z550_HK(LC) & B-F_HK(NLC)_26mths_37 °C	5.110293	0.000000
5	Z550_HK(NLC) & Z550_HK(LC)_26mths_37 °C	1.502584	0.132947
6	Z550_HK(NLC) & Z550_HK(NLC)_26mths_37 °C	1.520391	0.128414
7	Z550_HK(NLC) & B-F_HK(LC)_26mths_37 °C	5.500823	0.000000
8	Z550_HK(NLC) & B-F_HK(NLC)_26mths_37 °C	4.761776	0.000002
9	B-F_HK(LC) & Z550_HK(LC)_26mths_37 °C	5.893772	0.000000
10	B-F_HK(LC) & Z550_HK(NLC)_26mths_37 °C	5.943509	0.000000
11	B-F_HK(LC) & B-F_HK(LC)_26mths_37 °C	4.139920	0.000035
12	B-F_HK(LC) & B-F_HK(NLC)_26mths_37 °C	3.222693	0.001270
13	B-F_HK(NLC) & Z550_HK(LC)_26mths_37 °C	5.999019	0.000000
14	B-F_HK(NLC) & Z550_HK(NLC)_26mths_37 °C	5.670947	0.000000
15	B-F_HK(NLC) & B-F_HK(LC)_26mths_37 °C	4.670118	0.000003
16	B-F_HK(NLC) & B-F_HK(NLC)_26mths_37 °C	3.332350	0.000861

**Table 6 materials-16-02130-t006:** Changes in the microhardness of the test materials after aging in artificial saliva (AS) for 26 months. LC—light-cured surface of sample (the top side of a sample), NLC—non-light-cured surface (the bottom side of a sample), HK—Knoop hardness.

D	H/H_0_	1-H/H_0_	Material	Specimen Plane
0.02819	0.97181	0.02819	Z550	LC
0.13150	0.86850	0.13150	B-F	LC
0.04954	0.95046	0.04954	Z550	NLC
0.14779	0.85221	0.14779	B-F	NLC

## Data Availability

Data sharing is not applicable to this article.

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
