# Peer review of "Effect of Hydrothermal Factors on the Microhardness of Bulk-Fill and Nanohybrid Composites"

_materials, 2023, doi:10.3390/ma16052130_

Round 1

Reviewer 1 Report

I would like to thank the authors for the well written and well presented results. Some few comments to improve the manuscript:

-Add more recent papers to the introduction.

-Flow chart for the study for the study design. 

Author Response

The answer is in the attached file.

Reviewer 2 Report

The authors present an in vitro study evaluating the effect of hydrothermal factors on the microhardness of bulk-fill and nanohybrid composites. The topic is relevant to the journal's scope. However, the manuscript is hard to follow, and the experimental design and results need to be substantially improved.

Overall, the manuscript will benefit from language improvement. Several sentences need to shorten and simplified, and unnecessary words removed. See, for instance, lines 2-4 (you can just refer to bulk-fill and nanohybrid composites), 41 (remove systematically), 42-43 (remove the phrase), 45-46 (remove in the literature…), 92-93 (replace by this study aimed to), lines 290-291 (remove the sentence) and several more. Also, several information should be moved to other sections. This makes it difficult for the readers' comprehension and manuscript flow.

Title: I suggest replacing “simulated oral conditions” with “after thermocycling”. Oral conditions include other parameters, such as masticatory load, biofilm… which were not reproduced in this experiment.

Abstract: please define the meaning of the abbreviations before using them.

Lines 51-52: please clarify the sentence. Glass-based fillers are also inorganic fillers.

Lines 85-91: please move this information to the discussion section.

Lines 102-103 and 110-112: please move this information to the introduction or discussion sections.

Lines 115-122: this information should be moved to the introduction section after your study aim. This section is called materials and methods.

I recommend you use subtitles in your materials and methods section. For instance, “specimens preparation”, “Aging and thermocycling”, microhardness test, statistical analysis…

Section 2.2: how many specimens were prepared? What were the specimens’ dimensions and shape? How were they divided into groups? Please describe in detail all these parameters.

Section 2.2: please describe the preparation of the specimens.

Did you perform a sample size calculation? If not, why not?

Line 128: how many are half of the specimens? Please describe the n for each group and experiment.

Lines 129-130: please describe the “conditions simulating the oral cavity environment”.

Was the artificial saliva replaced during the one-month period?

Lines 133-134: what are the “conditioned specimens”?

The samples groups, duration of each experiment, microhardness evaluations… are incomprehensible. Please clarify this section. I suggest also adding an image (with the study group, time-points…) to facilitate the experimental design description.

Lines 157-159: which are the light-cured and non light-cured surfaces?

Materials and methods section: please add the description of the statistical analysis performed.

Figure 2, superior box: it shows two times LC, which I assume is light-cured. I also assume one of these refers to non light-cured. Please clarify.

Line 164: what is the null hypothesis? Is the hypothesis described as one in lines 115-116? Also, hypothesis one does not refer to comparing the surface mechanical properties of one material with another? Please clarify.

Lines 164-165: what comparison was made, and what statistical test was used to support your statement? What is the p-value?

Figures 1, 2, 3, 4, and 5 and tables 1, 2, 3 and 4: please describe the meaning of the abbreviation used in the figures/tables captions.

Figures 1, 2, 3, 4, and 5: why did you decide to present your results in such a format instead of presenting the mean and SD, for instance?

Lines 173-174: why did you use the Wilcoxon´s test? How was the data normality/not normality determined to choose the appropriate statistical test?

Why are there no results for B-F after 26 months?

I suggest figure 5, table 3 and related information be placed after figures 2 and 3 to allow an easier comparison between the two time-points.

Tables 2 and 4: please explain what you mean by H/H0 and 1-H/H0.

The discussion section needs to be rewritten. The authors fail to discuss their findings and provide a possible explanation for the differences between the two materials.

Author Response

The answer is in the attached file.

Reviewer 3 Report

1. Section 2. Materials and Methods: "The matrix of Z550 is composed of Bis-GMA, UDMA, Bis-EMA, PEGDMA and 101 TEGDMA resins" what was the ratio of matrix?

2. Did the author not use an initiator and accelerator in the matrix?

3. Add abbriviation list.

4. Section 2.2. Method; "The tests were carried out on light-cured (LC) and non-light-cured (NLC) surfaces of the specimens". How did authors make NLC composites?

5. Figure 3: why HK(LC) shows the maximum peak at 80-90.

6. Did the authors start this project 26 months ago?

7. Improved Introduction part and add some history of a dental composite, including recommended articles, to the reference list.

a. A comprehensive review: Physical, mechanical, and tribological characterization of dental resin composite materials. https://doi.org/10.1016/j.triboint.2022.108102

b. Tribological behavior of zinc oxide‐hydroxyapatite particulates filled dental restorative composite materials. https://doi.org/10.1002/pc.26597  

c. Comparative study of thermo-mechanical and thermogravimetric characterization of hybrid dental restorative composite materials. https://doi.org/10.1177/14644207211069763

Author Response

The answer is in the attached file.

Reviewer 4 Report

Manuscript materials-2150078 titled ‘Effect of hydrothermal factors on the microhardness of a bulk-fill composite and a nanohybrid composite under simulated oral conditions’ presented data on the effect of aging in artificial saliva and thermal shocks on the microhardness of the bulk fill composite compared to the nanohybrid composite. This topic is interesting, and authors have done a good job. Few of my comments and recommendations are as given below:

·      Title: Authors are advised to rephrase the title especially words ‘under simulated oral condition’ as this study presented data only after immersion in artificial saliva for one month.

·      A description of the detailed compositions of commercial products used in this study Filtek Z550 and Filtek Bulk-Fill resin composites would be useful for the readers

·      Line 115: Please be more specific in your first hypothesis ‘initial mechanical properties will be similar……’, is a very broad and generic statement.

·      Line 129: Please provide further details and reference to your study protocol, why did you select these parameters, a bit more details of artificial saliva would be more useful for the readers.

·      Line 154: Please indicate the sample dimensions of disk-shaped specimens used for microhardness tests. What was the thickness, why do you choose to tests microhardness on light-cured and non-light-cured surfaces of the specimens. What was your hypothesis in relation to this? However, you did not discussed the same variable in your discussion and conclusions.

·      Line 161-167 ‘results of the microhardness tests are summarized in histograms (Figs. 2–7)…….’, but you did not describe each histogram?

·      Would you please discuss more why microhardness decreased after thermocycling and is there is correlation with the other simulated aging conditions and variables?

Author Response

The answer is in the attached file.

Round 2

Reviewer 2 Report

The authors present an in vitro study evaluating the effect of hydrothermal factors on the microhardness of bulk-fill and nanohybrid composites. Some corrections were performed, but the manuscript is hard to follow and confusing.

First, several of my questions were not answered in the previous revision round. The authors paired some questions, and from that, they replied to just one of them. Please revise my previous questions and answer the ones you did not respond to.  

The manuscript needs significant language improvement. Several sentences need to shorten and simplified, and unnecessary words removed. For instance, in lines 23 and 24: you don’t need to write “abbreviated as X in this article”. Just write the abbreviation. Also, unnecessary words should be removed. For instance, 42 (remove systematically), 42-43 (academic and industrial centers), 46-46 (remove in the literature)…

Abstract, materials and methods section: this section is confusing. The authors refer to thermocycling after 1 month. Then they state that the control group was again put in the incubator, and the total aging time was 26 months, but it is unclear if something was done to the samples after this time. Please clarify this section.

Lines 49-50: please clarify the sentence's meaning.

Lines 53-54: please clarify the sentence. Glass-based fillers are also inorganic fillers. The authors stated this was corrected in the previous revision round, but it was not.

Lines 88-89: please clarify the sentence's meaning.

Study aim: I don’t see the need or the goal to state four hypotheses. So instead, I suggest you simplify your study aim.

Lines 114-124 and table 1 present the same information. Please choose one way to present the information and do not repeat it.

Table 1, materials type: if you state that 3M ESPE is a nanohybrid composite (particle size dimension), then you should apply the same classification to the bulk-fill resin and not say the type of this resin is “posterior restorative material”.

Table 1: please provide the meaning of the abbreviation on the table on a table caption.

Lines 134-135: please describe the specimens’ dimensions.

Lines 140-141: please describe your methods in sequence and not jump steps since this makes the manuscript hard to follow.

Section 2.2.2: please clarify this section. The authors first state that samples were aged for 1 month. Then the total time spent in the chamber was 26 months. Then, in line 145, they refer to “next, the conditioned specimens…”. What time is next? After 1 month? After 26 months?

Lines 174-175 and 178-180 present repeated information. Please remove the unnecessary information.

Lines 184-185: the normality was tested using the Shapiro-Wilk test. Please correct the sentence.

Lines 197-199: since you tested the normal/not normal distribution using a statistical test (the Shapiro-Wilk test), why do you state the graphs facilitate the assessment of the normality?

Figure 3, superior box: it shows two times LC, which I assume is light-cured. I also assume one of these refers to non light-cured. Please clarify.

Results refer to light-cured and non-light cured results, which are not explained in the materials and methods section. This needs to be described in the materials and methods section.

Figures and tables: please describe the meaning of the abbreviation used in the figures/tables captions. You need to describe in each caption what LC, NLC, HK.. is.

The results are confusing and hard to follow. It is unclear what the authors intended to compare. Is it the effect of light-curing or the effect of thermocycling? Please rearrange and clarify your results to make them easier to follow.

Did you perform a sample size calculation? If not, why not?

Figure 1 does not clarify the number of used specimens, as the authors state. Instead, it continues to refer to 50% of specimens. Please state the number of used specimens as number (n) in all the manuscript and not as a percentage. Also, the authors replied to several of my previous questions stating that figure 1 was added, but figure 1 does not answers my questions. Please add the necessary information and corrections to the manuscript instead of replying that figure 1 was added.

Author Response

The answer is in the attached file.

Round 3

Reviewer 2 Report

The authors present an in vitro study evaluating the effect of hydrothermal factors on the microhardness of bulk-fill and nanohybrid composites.

Abstract, line 24: the authors refer that samples placed in artificial saliva for one month are the control group. The control group for what? You tested two materials after being placed in artificial saliva for one month, two materials after thermocycling, and two materials after one month and 26 months of aging. So, I don’t understand this reference to a control group.  

Lines 47-48: as previously stated, glass-based fillers are also inorganic fillers. Your sentence needs to be corrected. You can refer to “70-89% of a glass-based filler and sometimes 20-30% of other inorganic nanofillers”.

Did you perform a sample size calculation? Sample size calculation refers to the number of specimens used and not to the size of the samples. Why did you decide to use 10 samples for each material?

Your sample size decreases to 5 per group after thermocycling and after 26 months of aging. So, you have half of your measurements. Do you consider this to have an impact on your results?

Figure 1: according to your figure, only half of the specimens were aged on artificial saliva, and the other half was readily thermocycled. This is opposite to what you describe in the text. Please correct.

Section 2.2.3: how many indentations were performed per sample?

Author Response

The answer is in the attached file.
